# A Novel DLG1 Variant in a Family with Brugada Syndrome: Clinical Characteristics and In Silico Analysis

**DOI:** 10.3390/genes14020427

**Published:** 2023-02-08

**Authors:** Maria d’Apolito, Francesco Santoro, Rosa Santacroce, Giorgia Cordisco, Ilaria Ragnatela, Girolamo D’Arienzo, Pier Luigi Pellegrino, Natale Daniele Brunetti, Maurizio Margaglione

**Affiliations:** 1Medical Genetics, Department of Clinical and Experimental Medicine, University of Foggia, 71122 Foggia, Italy; 2Department of Medical and Surgical Sciences, University of Foggia, 71122 Foggia, Italy; 3Cardiology Unit, Polyclinic Hospital of Foggia, 71122 Foggia, Italy

**Keywords:** Brugada syndrome, genetics, gene, DLG1

## Abstract

Background: Brugada syndrome (BrS) is an inherited primary channelopathy syndrome associated to sudden cardiac death. Overall, variants have been identified in eighteen genes encoding for ion channel subunits and seven genes for regulatory proteins. Recently, a missense variant in DLG1 has been found within a BrS phenotype-positive patient. DLG1 encodes for synapse associated protein 97 (SAP97), a protein characterized by the presence of multiple domains for protein–protein interactions including PDZ domains. In cardiomyocytes, SAP97 interacts with Nav1.5, a PDZ binding motif of SCN5A and others potassium channel subunits. Aim of the Study: To characterize the phenotype of an Italian family with BrS syndrome carrying a DLG1 variant. Methods: Clinical and genetic investigations were performed. Genetic testing was performed with whole-exome sequencing (WES) using the Illumina platform. According to the standard protocol, a variant found by WES was confirmed in all members of the family by bi-directional capillary Sanger resequencing. The effect of the variant was investigated by using in silico prediction of pathogenicity. Results: The index case was a 74-year-old man with spontaneous type 1 BrS ECG pattern that experienced syncope and underwent ICD implantation. WES of the index case, performed assuming a dominant mode of inheritance, identified a heterozygous variant, c.1556G>A (p.R519H), in the exon 15 of the DLG1 gene. In the pedigree investigation, 6 out of 12 family members had the variant. Carriers of the gene variant all had BrS ECG type 1 drug induced and showed heterogeneous cardiac phenotypes with two patients experiencing syncope during exercise and fever, respectively. The amino acid residue #519 lies near a PDZ domain and in silico analysis suggested a causal role for the variant. Modelling of the resulting protein structure predicted that the variant disrupts an H-bond and a likelihood of being pathogenic. As a consequence, it is likely that a conformational change affects protein functionality and the modulating role on ion channels. Conclusions: A DLG1 gene variant identified was associated with BrS. The variant could modify the formation of multichannel protein complexes, affecting ion channels to specific compartments in cardiomyocytes.

## 1. Introduction

Brugada syndrome (BrS) is an inherited primary channelopathy syndrome associated with sudden cardiac death. BrS, first described by Pedro and Josep Brugada in 1992, is an inherited disorder associated with the risk of ventricular fibrillation (VF) and sudden cardiac death in a structurally normal heart [1,2].

Diagnosis is based on a characteristic spontaneous electrocardiographic pattern (coved type ST-segment elevation ≥2 mm followed by a negative T-wave in one of the right precordial leads), associated with no other heart disease or on the presence of a type 1 BrS ECG induced by sodium channel blocker challenge among patients with no other heart disease who have survived a cardiac arrest [3].

Patients with BrS can present with syncope, seizures, and nocturnal agonal breathing due to polymorphic ventricular tachycardia (PVT) or VF. Both depolarization and repolarization abnormalities have been described in BrS [4,5].

BrS is believed to be a Mendelian disease with an autosomal dominant inheritance pattern with incomplete penetrance or variable expressivity [6,7,8,9,10]. However, the causal role of more than 20 genes, except SCN5A, has been recently disputed by the recent re-evaluation [11]. There is significant genetic heterogeneity underlying BrS. The majority of all pathogenic variants reported is loss-of-function mutations in SCN5A, the gene encoding for the α-subunit of the Na^+^ channel, frequently associated with a type 1 pattern [7]. To date, multiple pathogenic variants of 24 genes have been shown to alter the normal function of Na^+^, K^+^, Ca^2+^, and hyperpolarization-activated cyclic nucleotide-gated (HCN) channels, which mediate the ionic currents responsible for the cardiac action potentials [12,13]. Alterations of ion channel synthesis, membrane trafficking, and/or posttranslational modifications may lead to ion channel function defects that can give rise to genetically determined arrhythmogenic syndromes [14]. A group of proteins, expressed in cardiomyocytes, central to the trafficking of cardiac ion channels, belongs to the membrane-associated GUanylate kinase (MAGUK) family. These molecules are composed of different protein–protein interaction domains including three PDZ, one Src-homology-3 (SH3), and one guanylate kinase-like (GUK) domains [15]. SAP97 encoded by the DLG1 gene, the most well-characterized MAGUK protein in the myocardium, interacts with several ion channel families: KV1.5 [16,17], KV4.x [18,19], Kir2.x [20,21], and NaV1.5 [20,22]. Silencing of SAP97 in cardiomyocytes or in genetically modified mice significantly decreased IK1 [19,23]. Nav1.5 channels that generate the Na^+^ current (I_Na_) also bind to and co-localize with SAP97 at the intercalated disc [24]. SAP97 is a component of a macromolecular complex with the two critical channels underlying IK1 and INa and is involved in the formation of Kir2.1/NaV1.5 complexes. The reciprocal modulation between Nav1.5 and Kir2.1 and the respective ionic currents should be important in the ability of the heart to undergo self-sustaining cardiac rhythm disturbances [23]. A murine model for inducible, cardiac-targeted Sap97 ablation was generated to investigate arrhythmia susceptibility. This specific murine model demonstrated several ECG abnormalities, pronounced action potential prolongation subject to high incidence of arrhythmogenic after depolarizations, and notable alterations in the activity of the main cardiac ion channels [25].

In this paper, we present results of clinical and genetic investigations in an Italian family presenting with BrS. Genetic testing was performed with whole-exome sequencing (WES) using the Illumina platform. The effect of the identified gene variant was investigated by using in silico prediction of pathogenicity and within family linkage analysis.

## 2. Materials and Methods

### 2.1. Clinical and Genetic Investigation

We performed a clinical and genetic investigation of 13 family members of the proband. Diagnosis of BrS was established based on the 2022 ESC guidelines for the prevention of sudden cardiac death [3].

Clinical features such as age, sex, family history of sudden cardiac death (SCD), and pharmacological and electrical therapy were also assessed. Structural heart disease was excluded with conventional diagnostic methods including the treadmill test and echocardiogram.

Brugada type 1 ECG was defined as ST-segment elevation with coved type morphology of ≥2 mm in one or more right precordial leads in the fourth, third, and/or second intercostal space. Type 2 was defined as ≥2-mm J-point elevation, ≥1-mm ST-segment elevation, and a saddleback appearance, followed by a positive or biphasic T-wave in one or more right precordial leads. Type 3 was defined as having either a saddleback or coved appearance, but with an ST-segment elevation <1 mm in one or more right precordial leads [26].

All the patients with Brugada ECG pattern type 2 underwent pharmacological testing with flecainide intravenous with conversion to pattern 1 [27].

Clinical and genetic investigations were performed in accordance with the Helsinki declaration and based on written informed consent for clinical and genetic testing, and interventional treatment (if needed). Written informed consent was requested from their legal representatives for subjects under 18 years. All data presented in the manuscript were properly anonymized. The study was approved by the local ethical committee (n. 3261/2020).

### 2.2. Whole Exome Sequencing

Genetic testing was performed with whole-exome sequencing (WES) using the Illumina platform [28]. Genomic DNA was extracted from the whole blood of proband using standard methods. Enrichment was performed with Illumina DNA prep with Enrichment (San Diego, CA, USA) to capture all coding regions and exon-intron junctions (±50 bps) followed by Illumina NextSeq500 (San Diego, CA, USA). We obtained >99% of targeted regions covered more than 30 times and 99X mean depth. The raw data were then processed according to the Genome Analysis Toolkit (GATK 1.6) and were analysed using the software BaseSpace Variant Interpreter Annotation Engine 3.15.0.0 (Illumina). Variants were annotated according to the Human Genome Variation Society guidelines (HGVS), mapped to the human genome build GRCh37/UCSC hg19, and classified according to the criteria of the American College of Medical Genetics and Genomics [29]. Pathogenicity assessment for all rare genetic variants was performed according to ACMG2015 guidelines. To identify variants that were pathogenic, likely pathogenic or VUS, we looked up the variants’ minor allele frequency (MAF) in the Exome Aggregation Consortium (ExAC). We used 0.01 as an initial filtering criterion to limit the number of variants considered. In addition, further analysis was performed to identify variants associated with a given phenotype.

Exome analysis produced a large number of variants (12,154) of approximately 11,460 single nucleotide variations and approximately 666 indels. Variant annotation (i.e., exonic: intronic, and untranslated regions; exonic: synonymous, nonsynonymous, stop gain/loss, frameshift, allele frequency; and so on) and prioritization were performed with open-source software (Variant Interpreter, Illumina, San Diego, CA, USA).

Different approaches were used to minimize the number of potentially deleterious gene defects: (1) filtering based on a quality score of greater than 30; (2) excluding variants with a minor allele frequency of greater than 0.01 by comparison with the single nucleotide polymorphism database (http://www.ncbi.mln.nih.gov/snp (accessed on 18 November 2022)), Exome Aggregation Consortium (http://exac.broadinstitute.org/ (accessed on 18 November 2022)), Exome Variant Server (http://evs.gs.washington.edu/EVS (accessed on 18 November 2022), 1000 Genomes Projects (http://browser.1000genomes.org (accessed on 18 November 2022)), and published studies; (3) removing variants outside coding regions or synonymous coding variants; (4) selecting variants that segregate according to the presumed pattern of inheritance; and (5) querying disease databases, such as ClinVar (http://www.ncbi.nlm.nih.gov/clinvar (accessed on 18 November 2022)), OMIM, (http://www.omim.org (accessed on 18 November 2022)), and the Human Gene Mutation Database locus-specific database (http://www.hgvs.org/ (accessed on 18 November 2022)) to further prioritize the candidate gene variants. After the functional annotation step, which was undertaken based on effect on protein function and a priori knowledge of phenotype, approximately 116 variants were retained.

Virtual subpanels from these broad sequencing assays have been generated within the BaseSpace Variant Interpreter using a gene list associated to the disease (Appendix A). After this step, the number of variants were reduced to only 8. Of them, 7 resulted benign or likely benign according to ClinVar annotation, and only the DLG1 p.R519H remained to be further investigated.

### 2.3. Sanger Sequencing

According to the standard protocol [30], variants found by WES were confirmed in all members of the family by bi-directional capillary Sanger resequencing on the SeqStudio™ Genetic Analyzer System (Thermo Fisher Scientific, Waltham, MA, USA). Briefly, specific primers were designed to amplify exon 15 of DLG1 gene (NM_001204386.1). PCR products were then sequenced using the BigDye Terminator v.3.1 (Thermo Fisher Scientific, Waltham MA, USA).

### 2.4. In Silico Analysis and Protein Structure Modeling

To further prioritize the candidate gene variants, a functional annotation was undertook based on the effect on protein function and a priori knowledge of the phenotype. The in silico pathogenicity prediction to test the likely severity of a sequence alteration on protein function was performed using different bioinformatic tools, for nonsynonymous amino acid substitutions and for splicing variants. The effect of the p.R519H substitution was investigated by using in silico prediction of pathogenicity (PolyPhen-2, http://genetics.bwh.harvard.edu/pph2/ (accessed on 18 November 2022)); SIFT, http://sift.jcvi.org/siftbin/retrieve_enst.pl (accessed on 18 November 2022); and mutation assessor (http://mutationassessor.org/r3/ (accessed on 18 November 2022)). In addition, because these prediction tools tend to exploit a single information type (e.g., evolutionary conservation, structural, and biophysical properties) and/or are restricted to specific classes of mutations (e.g., to missense changes), we also used the CADD (combined annotation-dependent depletion) tool (http://cadd.gs.washington.edu/home (accessed on 18 November 2022)), a framework that integrates multiple annotations into one metric for scoring the deleteriousness of single nucleotide variants as well as insertion/deletions variants in the human genome (CADD https://cadd.gs.washington.edu/ (accessed on 18 November 2022)).

The DGL1 AlphaFold structure prediction model (AF-Q12959-F1-model_v4 (2).pdb) was then used as a template to investigate the putative pathogenic effect of the p.R519H substitution by using established available bioinformatics tools (MISSENSE3D and Swiss-PdbViewer). The DGL1 AlphaFold structure prediction model was used as the reference sequence UniProt Q12959.

## 3. Results

### 3.1. Proband Characteristics

The proband was a 74-year-old man who was admitted to the emergency department because of syncope at rest. The 12-lead rest electrocardiogram (ECG) showed a type 1 Brugada pattern, first-degree atrioventricular (AV) block, right bundle branch block and left posterior hemiblock (Figure 1). The 24 h Holter showed a paroxysmal third-degree complete AV block.

Echocardiography demonstrated normal left ventricular systolic function with slight impairment of left ventricular global longitudinal strain (LV-GLS) (−15%), a sigmoid-shaped interventricular septum, and enlargement of the right ventricle. Besides cardiac lesions, the proband had history of hypertension and deep venous thrombosis. During hospitalization, a dual-chamber cardiac defibrillator was implanted.

### 3.2. Whole-Exome Sequencing of Brugada Syndrome Patients Revealed Variant in Gene Known for Their Association with the Disease

WES of the index case, performed assuming a dominant mode of inheritance, identified a heterozygous variant in the exon 15 of the DLG1 gene. No variants involving other known BrS-susceptibility genes were detected (see Appendix A). The variation identified was a nucleotide substitution c.1556G>A, predicting a missense amino acid change involving the substitution of an arginine by a histidine p.(R519H). The NCBI GenBank accession number NM_001366207.1 and UniProt identifier Q12959-4 were used as the reference sequence. This variant (rs141544348) is classified as VUS (a variant of unknown significance, Class III) according to ACMG (2015) recommendations. The variant identified is rare, being reported in GnomAD with an allele frequency of 3.62 × 10^−4^. PolyPhen-2, SIFT, combined annotation-dependent depletion (CADD) scores were calculated to evaluate the pathogenicity of the identified variation. Findings from in silico analyses indicated the possibility that the variant could be deleterious (Table 1).

According to the standard protocol, the variant R519H found by WES was confirmed in the index case and investigated in all members of the family by direct capillary Sanger sequencing.

Sequence conservative analysis revealed that the sequences of DGL1 exhibited a high conservation among various species. (Figure 2). The amino acid residue #519 lies near the end of the PDZ3 domain. The DLG1 AlphaFold structure prediction model (AF-Q12959-F1-model_v4 (2).pdb) was used as a template to investigate the putative pathogenic effect of the p.R519H variant. The in silico analysis (MISSENSE3D; http://www.sbg.bio.ic.ac.uk/missense3d/ (accessed on 18 November 2022)) detected a wild-type salt bridge between the NH1 atom of R519 and the OE1 atom of E516 (distance: 4.990 Å). The substitution with a histidine was predicted to disrupt this bond. Swiss-Pdb Viewer modeling shows comparison of the predicted structures of both wild-type and mutant protein (https://spdbv.unil.ch/ (accessed on 18 November 2022)). The wild-type residue is involved in side-chain/main-chain H-bonds (Figure 3).

### 3.3. Pedigree Investigation

All proband’s sons (II-1, II-2, II-3, II-4) had a BrS type 1 ECG drug induced with flecainide testing, and direct sequencing revealed the DLG1 R519H variant in all of them. The evaluation of second-degree relatives identified two additional family members (III-4, III-7) carrying the same substitution and a BrS type 1 ECG drug induced with flecainide testing. (Figure 4).

As a whole, seven genotype-positive carriers and six negative noncarriers were investigated. Among the carriers, three had history of syncope (I, II-5, III-4). The proband experienced it at rest (I-1), one during exercise (II-5), and one during fever (III-4). The patient with syncope during exercise (II-5) underwent first the treadmill test and then electrophysiological study and electrical programmed stimulation. Both tests were normal, and no ventricular arrhythmias were recorded or induced (Table 2).

ECG evaluation showed low QRS voltage in all peripheral leads in a patient (II-5, Appendix A), deep S-wave in lead I (I-1, Figure 1).

Among genotype-positive carriers, at echocardiographic evaluation, one had slight impairment of LV-GLS (-16.5%) (II-7) and one had mitral valve prolapse of the left anterior leaflet (III-4) (Table 2).

All noncarriers showed a normal resting ECG and echocardiographic examination was normal.

A LOD score was calculated for quantifying the segregation evidence for the DLG1 p.R519H variant. A LOD score value of 2.5 was obtained.

## 4. Discussion

We report in a large multigenerational family with Brugada syndrome a novel DLG1 gene variant that was associated with typical electrocardiographic presentation and heterogenous cardiac phenotype.

### 4.1. Genes and BrS

The inheritance of BrS was believed to present an autosomal-dominant mode of transmission with incomplete penetrance. The penetrance of BrS is age- and sex-dependent, and most lethal events occur in men after the fourth decade of life. In a long-term follow-up study, age-related depolarization abnormalities (e.g., slowing of conduction) in the heart modulated the clinical ECG phenotype, especially in patients with SCN5A-positive pathogenic variants [31]. Gender is also a modifier of both phenotype and the risk of SCD in BrS. The relative risk for cardiac events in men is 3.34 times that in women. The sex-specific differences may be linked to the differential effect of sex hormones on cardiac ion channel current densities and functional cardiac repolarization reserves. In the past decades, more observations suggest that BrS has a heterogeneous genetic basis and is a disease with a more complex inheritance. Overall, 40% of BrS cases are familial, whereas other cases are sporadic. Despite its symptomatic or asymptomatic status, a positive family history of sudden cardiac death at a young age (45 years) in BrS patients was quite low (10–30%). In addition, low disease penetrance was observed based on ECG analysis in family members carrying SCN5A pathogenic variants (16%) [32].

Familial linkage analyses have largely been unsuccessful in uncovering new disease-causing genes because most of the BrS families are rather small for powerful linkage analysis, low penetrance, and variable expressivity of diseases. The current evidence shows that the genetic architecture of BrS is complex, and it is more like an oligogenic disease [33].

Indeed, the current expert consensus statement recommends mutation-specific genetic testing for family members and appropriate relatives following identification of the BrS causative gene variants in an index patient (Class I recommendation). The primary role of genetic testing in BrS is to confirm the disorder in the index patient, and cascade testing in relatives to distinguish those who need clinical follow-up (for development of conduction disease or syncopal episodes) and preventive measures (e.g., avoiding specific drugs, prompt management of fever) from those who are both clinically and genetically unaffected. At the state of the art, the result of genetic testing is not considered one of the criteria for BrS diagnosis [3]. Genetic testing is not recommended in the absence of a diagnostic ECG. Nevertheless, targeted genetic testing can be useful for any patient in whom a cardiologist has established a clinical index of suspicion for BrS based on clinical examination (patient’s clinical history, family history, and ECG phenotype). In addition, pathogenic variant-specific genetic testing in family members of successfully genotyped probands may play a decisive role in determining who should take precautions in certain conditions and who should be followed-up.

### 4.2. Ion Channels and BrS

Since decades, it is well known that variants in genes encoding for ion channels are involved in the pathogenesis of arrhythmogenic diseases. Only recently, genes encoding for modulators of the expression, functionality, and cellular distribution of ion channels has been demonstrated to play a significant role in cardiac excitability.

Many variants related to BrS involve genes which are important in regulating sodium channel function. SCN5A plays a key role in the rapid depolarization during the phase 0 of the cardiac action potential through a loss of function of a sodium channel. Therefore, a pathogenetic variant in this gene can bring about a slower impulse conduction. A large number of SCN5A pathogenic variants have been described and this number is destined to grow. Loss of function variants in genes encoding for β subunits of Nav 1.5 channel (SCN1B, SCN2B, and SCN3B) have been suggested to play a role in BrS.

Calcium channel BrS-susceptibility genes were found among calcium channels (CACNA1C, CACNB2b, and CACNA2D1).

Gain-of-function variants associated with enhanced K+-channel function have also been described in BrS [34]. These pathogenic variants are able to shorten the cardiac action-potential duration; therefore, provide a substrate for re-entrant arrhythmias [35].

Potassium channels, apart from sodium and calcium channels, that provide putative gain of function variants in genes encoding channels that conduct outward potassium currents (*KCND3, KCNE3, KCNE5*, and *KCNJ8*) have also been reported in a few BrS cases.

### 4.3. Ion Channel-Interacting Proteins and BrS

Variants in genes encoding for ion channel-interacting proteins have been associated with the occurrence of arrhythmogenic diseases. A decreased INa has been associated with a variant of the sodium channel-associated RAN guanine nucleotide release factor (RANGRF) gene, which encodes MOG1, involved in the carriage of Nav 1.5 to the membrane [36]. Pathogenic variants in the glycerol-3-phosphatedehydrogenase1-like (GPD1L) gene may also affect the trafficking of cardiac Naþ channels to the cell surface and cause 50% reduction in the inward Naþ current.

### 4.4. DLG1 Gene Encoding SAP97 Protein

DLG1, a gene encoding for SAP97, a protein involved in the functional regulation of ion channels, has been previously identified in BrS patients [25]. SAP97 plays an important role in the mechanisms underlying channel localization and clustering of potassium ion channels [18]. Ion channel proteins bind to PDZ domains of SAP97 protein by their C terminus, in agreement with the binding selectivity of PDZ domains. SAP97 is highly expressed in myocardium and regulates the targeting of cardiac potassium ion channels in the sarcolemma.

SAP97 interacts with several ion channels mainly involved in the sodium and potassium current and is involved in the formation of Kir2.1/NaV1.5 complexes. Sodium current changes have been described in BrS due to variants in the SCN5A gene that encodes the cardiac sodium channel Nav1.5. The BrS ECG phenotype is most often the result of loss of function in Nav1.5. Indeed, BrS typically manifests in vitro as a loss of whole-cell sodium currents and in vivo as conduction slowing 

A knockout murine model for inducible, cardiac-targeted Sap97 ablation, was used to investigate arrhythmia susceptibility [25]. This specific murine model showed that a DLG1 variant was associated with a longer action potential duration (APD) and a decrease for Ito and IK1 current, respectively. These changes could be associated to a higher incidence of arrhythmias after depolarizations. In the same study, a different pathogenic variant was found in two patients (proband and his daughter) with BrS and history of SCD but not in a cohort of patients with long QT syndrome. The identified DLG1 variant in this study was c.2480T>C (g.196786760A>G) resulting in a M827T missense amino acid substitution at a residue localizing to the guanylate kinase-like (GUK) domain of this protein. The nucleotide affected occurred at the very last nucleotide position of exon 23 of the protein and could eventually result into an abnormal splicing. Cells expressing the identified variant showed characteristics indicating a significant ADP shortening [25]. Results strongly suggest that DLG1 loss-of-function variants are associated with long QT syndrome, whereas gain-of-function ones with BrS.

In patients presenting with BrS, only another DLG1 variant has been suggested to be associated [37]. Indeed, a DLG1 common variant (rs34492126) has been showed in a man and two sisters with BrS. This polymorphism is featured by a marked increase of the Ito current and shortens the QT interval and the APD and might contribute to susceptibility to BrS [38].

### 4.5. DLG1 Gene p.R519H Variant

In this present report, seven variant carriers were identified with similar electrocardiographic presentation (mainly BrS ECG pattern type 1 drug induced) but with an heterogenous cardiac phenotype. Only three out of seven experienced syncope and two out of seven had some echocardiographic abnormal findings. Although no experimental model was tested to support the pathogenicity of the DLG1 variant identified, a series of findings suggest a role for the DLG1 p.R519H missense substitution. The variant identified affected an amino acid residue highly conserved among all species investigated. It occurred in an α-helix located in close proximity of a PDZ domain. The substitution of a histidine for an arginine was predicted to disrupt an H-bond. As a consequence, it is likely that a conformational change affects protein functionality and the modulating role on ion channels.

An intriguing and independent suggestion is provided by the detection of the variant in all family members who presented an electrocardiographic pattern suggestive of BrS. Indeed, over three generations all of p.R519H carriers had similar electrocardiographic changes. The use of segregation studies in which family members are genotyped to determine if a variant co-segregates with a disease can be a powerful piece of evidence to suggest a meaningful association. The individual LOD score generated suggested that the pattern of phenotypes within the family investigated is consistent with the transmission of a major gene for that phenotype.

However, this analysis provides initial evidence, but not a definitive proof, that a DLG1 pathogenic variant has a major effect on a BrS phenotype. Indeed, LOD score analysis assumes that the trait being analysed is a single-locus Mendelian phenotype. Evidence from this kind of analysis can lend support for a genetic basis in disease aetiology but cannot indicate the number of genes involved in susceptibility or the magnitude of their effects. The interpretation of genetic alterations and their translation into clinical practice is one of the main challenges in modern clinical genetics. In BrS, family segregation investigation and a comprehensive genotype–phenotype correlation help to unravel the role of genetic variants; however, incomplete penetrance and variable expressivity make unclear their definitive roles.

The frequency of the variant in the general population is rare, being reported in GnomAD with an allele frequency of 3.62 × 10^−4^, 6.0 × 10^−4^ in Europeans. However, it is higher than expected for a dominant Mendelian allele with a penetrance of 0.5 (6 × 10^−6^). This finding disagrees with the relatively high penetrance suggested by pedigree data. The co-segregation of other genetic factors in the family might explain this discrepancy. Alternatively, the variant could act as a susceptibility, non-Mendelian, risk factor.

## 5. Limitations

No experimental model was performed. The present study evaluated only an informative family and results cannot be generalized. 

## 6. Conclusions

A DLG1 gene variant was identified to be associated with Brugada syndrome. A series of clinical and laboratory findings provides initial evidence, but not a definitive proof, that the DLG1 p.R519H variant has a major effect on a BrS phenotype. This variant could modify the formation of multichannel protein complexes, affecting ion channels to specific compartments in cardiomyocytes.

## Figures and Tables

**Figure 1 genes-14-00427-f001:**
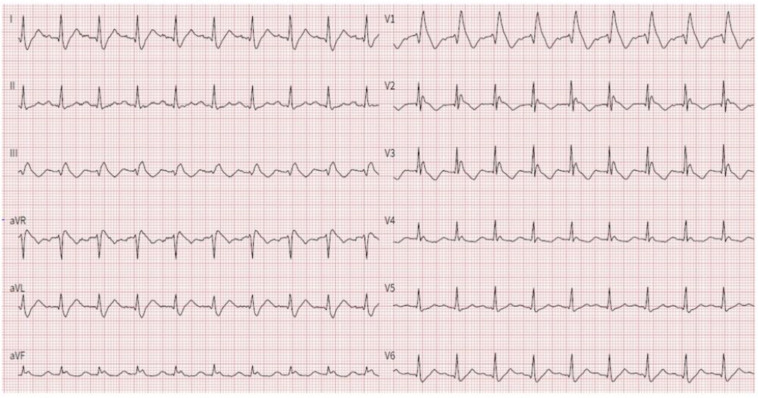
Rest 12-lead ECG of the proband during admission at emergency department for syncope, showing Brugada type 1 ECG pattern.

**Figure 2 genes-14-00427-f002:**
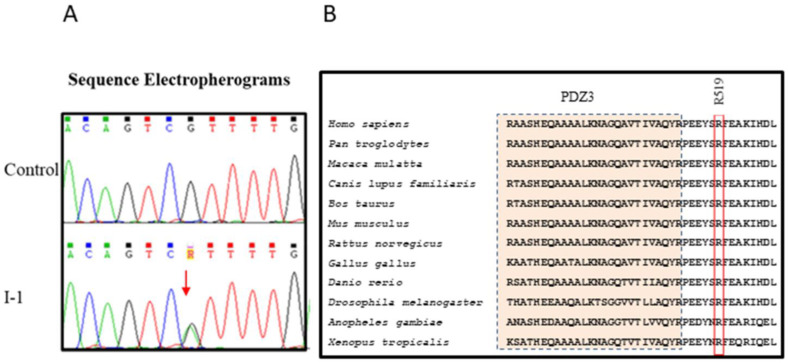
(**A**) Sanger sequencing showing the c.1556G>A substitution in the propositus. (**B**) Alignment of a portion of the PDZ3 domain and the amino acid position #519 (highlighted in red) along different species.

**Figure 3 genes-14-00427-f003:**
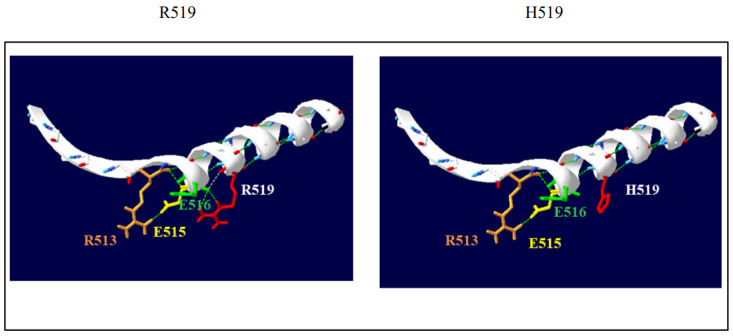
Comparison of the predicted structures of both wild-type (R519) and mutant protein (H519) using SwissPdbViewer (https://spdbv.unil.ch/ (accessed on 18 November 2022)); H-bonds are displayed as green dotted lines in the mutant protein.

**Figure 4 genes-14-00427-f004:**
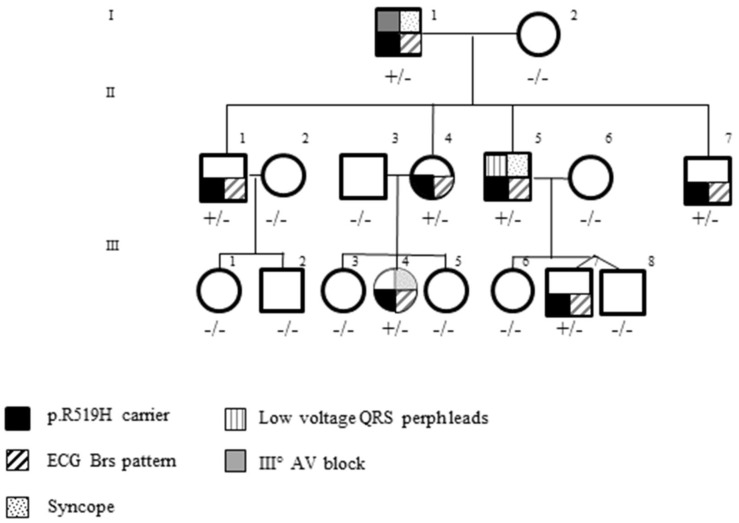
Pedigree of the proband carrying the DLG1 variant. Circles indicate females, squares males. Open symbols represent asymptomatic individuals, filled symbols indicate carriers of the DLG1 variant, the presence of a BrS type 1 ECG and an arrhythmogenic syndrome according to the trauma code. +/−: heterozygote for p.R519H variant; −/−: non carrier.

**Table 1 genes-14-00427-t001:** In silico predicted pathogenicity of the DLG1 variant.

Gene	Nucleotide Change	Amino Acid Change	PolyPhen-2	SIFT	CADD
DGL1	CGT/CAT	R519H	1	0	31

**Table 2 genes-14-00427-t002:** Cardiac phenotypes of the members carrying the DLG1 variant.

Patient	Sex	Age	History of Syncope	ECG Features	Echocardiogram Features
I	M	74	Yes	Spontaneous type 1 BrS pattern; III AVB	Impaired LV-GLS; Sigmoid-shaped IV septum
II-1	M	54	No	Type 2 BrS pattern	--
II-4	F	48	No	Type 2 BrS pattern	--
II-5	M	47	Yes	Type 2 BrS pattern Low-voltage peripheric limbs	Impaired LV-GLS
II-6	M	43	No	Type 2 BrS pattern	--
III-4	F	20	Yes	Type 2 BrS pattern	Mitral valve prolapse
III-7	M	11	No	Type 2 BrS pattern	--

Legend: AVB = atrio-ventricular block; BrS = Brugada syndrome; LV-GLS = left ventricle global longitudinal strain.

## Data Availability

Data available on request from the corresponding authors, due to restrictions on patient’s privacy as established in the informed consent.

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
