# Peer review of "A Novel DLG1 Variant in a Family with Brugada Syndrome: Clinical Characteristics and In Silico Analysis"

_genes, 2023, doi:10.3390/genes14020427_

Round 1

Reviewer 1 Report

This is an interesting study in a family with Brugada syndrome. 

My only comment relates to the ECG of the proband which is shown in figure 1. In addition to the ST elevation and borderline first degree AV block one can observe that the patient has a complete right bundle branch block (QRS 160 ms) and a left posterior hemiblock (thus a trifascicular block), something not mentioned in the text. It is not surprising, therefore, that the patient showed complete AV block later on. 

Author Response

Major points

Point 1: My only comment relates to the ECG of the proband which is shown in figure 1. In addition to the ST elevation and borderline first degree AV block one can observe that the patient has a complete right bundle branch block (QRS 160 ms) and a left posterior hemiblock (thus a trifascicular block), something not mentioned in the text. It is not surprising, therefore, that the patient showed complete AV block later on. 

Response 1: we thanks the Reviewer for the helpful comment. "As kindly suggested, ECG features of the proband have been added in the result section, "proband characteristics" (results, lines 179-183).

Reviewer 2 Report

This study describes a novel variant in the DLG1 gene in a reasonably large family pedigree which seems to segregate with a Brugada phenotype. This is a plausible disease gene and the report builds on an earlier paper from Musa et al describing a different variant in this gene in a Brugada patient. 

However, there are significant limitations with the study as currently presented. The rate of validation of proposed Brugada genes has historically been very low (with all but SCN5A considered disputed by the recent ClinGen re-evaluation), which means that robust methods need to be applied for any novel associations. One of the major issues with this association is the frequency of the variant in the population - although the authors claim it is rare in gnomAD, it is actually far too common to be a reasonably penetrant and Mendelian-like variant for Brugada syndrome. The European MAF (0.0006) is actually even higher than the overall MAF described in the Results, far greater than the proposed threshold for a penetrant variant (see PMID:28518168). It is still possible of course that the variant could act as a non-Mendelian risk factor, but strong functional or statistical genetic evidence would be needed to validate this. More discussion of this frequency, and the consequences for the interpretation of the potential effect of this variant, needs to be added. And the conclusions about the potential pathogenicity of the variant should be modified accordingly. 

Far too little detail on the analysis of the WES data is provided. More than one rare variant would have identified in the patient, but no information is provided about the other variants detected or how and why the DLG1 variant was prioritised for analysis. 

The computational analysis of the variant described can of course provide some supporting data for pathogenicity, but this should be interpreted more cautiously given the non-specific nature of these tools. Descriptions like "supported a causal role for the mutation" and "a likelihood or causing disease" should be moderated. 

For the pedigree analysis - were any clinical studies performed on the non-carriers? Figure 4 should more clearly indicate the non-carriers (+/- or -/- would be better annotation for the genetic status than filled segments of the squares and circles). The third generation is incorrectly labelled (II). A LOD score should be provided for quantifying the segregation evidence for this variant. 

Minor points:

The term "variant" or "pathogenic variant" is preferable to "mutation".

The description of BrS as a Mendelian disease is not particularly accurate - although rare variants in SCN5A are found in about 20% of cases, there are examples of non-segregation in such families and a large contribution of common variants. It would be more accurate to describe the genetic aetiology as complex with certain features of Mendelian disease.

The discussion of ref 25 (Musa et al) refers to the same variant being studied in mice and detected in a Brugada patient. Isn't the mouse data there related to a knockout genotype? 

Common variants in candidate genes can provide supporting evidence for a disease role (there is often overlap between Mendelian genes and GWAS loci). However, the distinction between these variant types need to be appropriately described. As the P888L variant is common, the fact that it was found to be present in 3 family members with Brugada offers little evidence for association. The functional data described in Ref 34 is more relevant but as a candidate SNP study, it should be interpreted with caution. 

Author Response

Major points

Point 1: One of the major issues with this association is the frequency of the variant in the population - although the authors claim it is rare in gnomAD, it is actually far too common to be a reasonably penetrant and Mendelian-like variant for Brugada syndrome. The European MAF (0.0006) is actually even higher than the overall MAF described in the Results, far greater than the proposed threshold for a penetrant variant (see PMID:28518168). It is still possible of course that the variant could act as a non-Mendelian risk factor, but strong functional or statistical genetic evidence would be needed to validate this. More discussion of this frequency, and the consequences for the interpretation of the potential effect of this variant, needs to be added. And the conclusions about the potential pathogenicity of the variant should be modified accordingly. 

Response 1: we thanks the Reviewer for the helpful comment. According to his suggestion, the Discussion section has been revised and the interpretation of the potential effect of this variant has been argumented (lines 387-400).

Point 2: Far too little detail on the analysis of the WES data is provided. More than one rare variant would have identified in the patient, but no information is provided about the other variants detected or how and why the DLG1 variant was prioritised for analysis. 

Response 2: more details on the analysis of the WES were provided and the Materials and Method section has been changed accordingly (lines 126-149). 

Point 3: The computational analysis of the variant described can of course provide some supporting data for pathogenicity, but this should be interpreted more cautiously given the non-specific nature of these tools. Descriptions like "supported a causal role for the mutation" and "a likelihood or causing disease" should be moderated. 

Response 3: we agree with the Reviewer’s comment and revised the whole text accordingly (see Abstract lines 30-35; Discussion, and Conclusions lines 405-409). 

Point 4: For the pedigree analysis - were any clinical studies performed on the non-carriers? Figure 4 should more clearly indicate the non-carriers (+/- or -/- would be better annotation for the genetic status than filled segments of the squares and circles). The third generation is incorrectly labelled (II). A LOD score should be provided for quantifying the segregation evidence for this variant.

Response 4: all noncarriers underwent resting ECG and echocardiographic examination. This information has been included in the present version of the manuscript (Result, lines 254-255). The Figure 4 has been changed as requested and now “+/-“ or “-/-“ notation are included. The third generation has been correctly labelled. A LOD score analysis was performed and data are now presented (Results, lines 256-257).

Minor points

Point 1: The term "variant" or "pathogenic variant" is preferable to "mutation".

Response 1: we agree with the Reviewer’s comment and have revised the whole text accordingly.

Point 2: The description of BrS as a Mendelian disease is not particularly accurate - although rare variants in SCN5A are found in about 20% of cases, there are examples of non-segregation in such families and a large contribution of common variants. It would be more accurate to describe the genetic aetiology as complex with certain features of Mendelian disease.

Response 2: we agree with the Reviewer’s comment and the complexity of the BrS inheritance has been clearly discussed (Discussion, lines 267-286).

Point 3: The discussion of ref 25 (Musa et al) refers to the same variant being studied in mice and detected in a Brugada patient. Isn't the mouse data there related to a knockout genotype? 

Response 3: we recognize that data from ref#25 were poorly presented. In the present version of the manuscript results have been clearly indicated (Discussion, lines 347-360).

Point 4: Common variants in candidate genes can provide supporting evidence for a disease role (there is often overlap between Mendelian genes and GWAS loci). However, the distinction between these variant types need to be appropriately described. As the P888L variant is common, the fact that it was found to be present in 3 family members with Brugada offers little evidence for association. The functional data described in Ref 34 is more relevant but as a candidate SNP study, it should be interpreted with caution. 

Response 4: we agree with the Reviewer’s comment and data from ref#34 (now #39) are more clearly presented (Discussion, lines 361-365)

Round 2

Reviewer 2 Report

I thank the authors for their extensive responses to the reviewer comments. However, the information provided about the WES filtering pipeline is still vague and incomplete.

The filtering pipeline (step 4) states:

"filtering the data for novelty by comparison with the Single Nucleotide Polymorphism Database (http://www.ncbi.mln.nih.gov/snp), Exome Aggregation Consortium (http://exac. broadinstitute.org/), Exome Variant Server (http://evs.gs.washington.edu/EVS), 1000 Genomes Projects (http://browser.1000genomes.org), and published studies;"

As the variant is present in dbSNP, ExAC etc, then it cannot be novel compared to these resources. In any case, filtering for complete novelty in such databases can remove relevant variants. A single filtering step by frequency is more appropriate to remove variants too common to be causal.

Filtering step 5 ("querying disease databases, such as ClinVar (http://www.ncbi.nlm.nih.gov/clinvar), OMIM, (http://www.omim.org), and the Human Gene Mutation Database Locus-specific database (http://www.hgvs.org/).") is far too vague - what does querying these databases mean? Were only variants previously implicated in disease retained? 

The frequency of the variant in European population is briefly mentioned in the last lines of the Discussion but rather glossed over. The pedigree data suggests a variant of relatively high penetrance but the population data would argue against this. More discussion on these points is warranted as to what might explain this discrepancy - e.g. other shared genetic factors in the family? 

The manuscript would also benefit from careful re-reading and editing to improve the clarity of the language used. For example:

Line 244 - "After these filtering strategies, approximately 116."

Discussion - "DLG1, a gene encoding for SAP97" followed in the next paragraph by "SAP97, encoded by the DLG1 gene".

Author Response

Response to Reviewer 2 Comments

Minor points

Point 1: The filtering pipeline (step 4) states:

"filtering the data for novelty by comparison with the Single Nucleotide Polymorphism Database (http://www.ncbi.mln.nih.gov/snp), Exome Aggregation Consortium (http://exac. broadinstitute.org/), Exome Variant Server (http://evs.gs.washington.edu/EVS), 1000 Genomes Projects (http://browser.1000genomes.org), and published studies;"

As the variant is present in dbSNP, ExAC etc, then it cannot be novel compared to these resources. In any case, filtering for complete novelty in such databases can remove relevant variants. A single filtering step by frequency is more appropriate to remove variants too common to be causal. 

Response 1: we thanks the Reviewer for the helpful comment. We recognoze that as formulated, the text could be unclear for the reader. According to his suggestion, the Materials and Methods section has been revised and the interpretation of the potential effect of this variant has been argumented (lines 132-137).

Point 2 Filtering step 5 ("querying disease databases, such as ClinVar (http://www.ncbi.nlm.nih.gov/clinvar), OMIM, (http://www.omim.org), and the Human Gene Mutation Database Locus-specific database (http://www.hgvs.org/).") is far too vague - what does querying these databases mean? Were only variants previously implicated in disease retained? 

Response 2: more details on the Filtering step 5 analysis of the WES were provided and the Materials and Method section has been changed accordingly (lines 139-143). 

Point 3: The frequency of the variant in European population is briefly mentioned in the last lines of the Discussion but rather glossed over. The pedigree data suggests a variant of relatively high penetrance but the population data would argue against this. More discussion on these points is warranted as to what might explain this discrepancy - e.g. other shared genetic factors in the family? 

Response 3: we thanks the Reviewer for the helpful comment and revised the Discussion section accordingly (lines 396-401). 

Point 4: The manuscript would also benefit from careful re-reading and editing to improve the clarity of the language used.

Response 4: the text has been revised and the indicated sentences amended.